# Genome-Wide Association Study Identifies Potential Regulatory Loci and Pathways Related to Buffalo Reproductive Traits

**DOI:** 10.3390/genes16040422

**Published:** 2025-03-31

**Authors:** Wangchang Li, Qiyang Xie, Haiying Zheng, Anqin Duan, Liqing Huang, Chao Feng, Jianghua Shang, Chunyan Yang

**Affiliations:** 1Laboratory of Buffalo Genetics, Reproduction and Breeding, Guangxi Buffalo Research Institute, Chinese Academy of Agricultural Sciences, Nanning 530001, China; liwangchang1019@163.com (W.L.); 2218391048@st.gxu.edu.cn (Q.X.);; 2Laboratory of Animal Breeding, Disease Control and Prevention, College of Animal Science & Technology, Guangxi University, Nanning 530004, China

**Keywords:** buffalo, reproductive traits, genome-wide association study

## Abstract

Background: The reproductive performance of water buffalo significantly impacts the economic aspects of production. Traditional breeding methods are constrained by low heritability and numerous influencing factors, making it difficult to effectively improve reproductive efficiency. Genome-wide association studies (GWAS) offer new possibilities for exploring reproductive traits in water buffalo, opening up new avenues for efficient breeding. Methods: Using whole-genome resequencing, we identified quantitative trait loci (QTLs) associated with four suggestive reproductive traits: calving interval (CI), calf birth weight (CBW), dam birth weight (BW), and age at first calving (FCA). The study focused on identifying genetic variants that influence these reproductive traits. Results: Our research identified 52 suggestive regulatory loci associated with reproductive traits in water buffalo. Based on a 50 kb interval, we annotated these loci to 58 candidate genes. These loci involve genes such as *AGBL4*, *GRM1*, *NCKAP5*, and *NRXN1*, which are primarily enriched in pathways including the FOXO signaling pathway, calcium ion pathways, estrogen signaling pathway, and phospholipase D signaling pathway. These pathways directly or indirectly regulate the reproductive efficiency of water buffalo. Conclusions: This study has revealed suggestive regulatory genes (*AGBL4*, *GRM1*, *NCKAP5*, *NRXN1*) associated with reproductive traits in water buffalo. This not only enhances our understanding of the molecular mechanisms underlying complex traits but also points towards strategies for improving the reproductive capacity of water buffalo. These findings provide a solid foundation for future breeding programs aimed at enhancing water buffalo productivity.

## 1. Introduction

Water buffalo plays a pivotal role in human agriculture, serving not only as a robust force for cultivation but also yielding essential products such as meat, milk, leather, and fertilizer. Reproductive traits in water buffalo, representing vital economic factors, are intricate quantitative traits. Traditional breeding methods for enhancing reproductive qualities in water buffalo have been hindered by low heritability and various influencing factors. The advent of whole-genome association analysis (GWAS) has opened new avenues for exploring these quantitative traits [1]. GWAS methods have unearthed numerous single nucleotide polymorphisms (SNPs) and quantitative trait loci (QTLs) linked to the growth, reproduction, and meat quality traits of livestock and poultry [2,3,4,5]. Over the past two decades (2000–2020), GWAS has identified a multitude of QTLs associated with reproductive traits in both cattle and water buffalo [6].

The water buffalo genome assembly forms the cornerstone for genome-wide association studies (GWAS). Since the unveiling of the human genome in 2001, there has been significant progress in genomics, paralleled by advancements in high-throughput sequencing technologies that have lowered sequencing costs, enabling broader species sequencing. Reference genomes for various species, including chicken [7], cattle [8], pigs [9], sheep [10], ducks [11], and geese [12], have emerged successively.

GWAS is a powerful tool used to identify genomic regions associated with specific traits. By genotyping large numbers of individuals and correlating this data with phenotypic information, researchers can pinpoint genes and loci linked to reproductive traits such as calving interval, birth weight, and age at first calving. For example, researchers conducted a genome-wide association study (GWAS) on reproductive traits in Hanwoo cattle, focusing on age at first calving (AFC), calving interval (CI), gestation length (GL), and number of artificial inseminations per conception (NAIPC). Through GWAS, the researcher identified 10 candidate genes associated with age at first calving (AFC) [13]. Several studies have utilized genome-wide association studies (GWAS) to explore potential genetic factors influencing reproductive traits such as oocyte and embryo numbers. These studies have identified the protein-coding genes *ARNT*, *EGR1*, *HIF1A*, *AHR*, and *PAX2* as strong markers for oocyte and embryo production in Gir cattle. The discovery of these genes provides important insights into the molecular mechanisms underlying reproductive performance in Gir cattle and offers scientific support for optimizing reproductive management strategies [14,15].

Our study elucidates the regulatory mechanisms underlying reproductive traits in water buffalo. Through whole-genome resequencing and association analysis of reproductive traits in water buffalo, we have identified suggestive molecular loci (*AGBL4*, *GRM1*, *NCKAP5*, and *NRXN1*) closely associated with these traits. Further investigation revealed that these candidate genes are primarily enriched in pathways such as the FOXO signaling pathway, estrogen signaling pathway, and calcium ion pathways. These findings provide effective support for enhancing assisted breeding programs aimed at improving water buffalo reproduction and offer a guide for future scientific breeding strategies.

## 2. Materials and Methods

### 2.1. Phenotypes and Animal Resources

This study utilized data collected from 120 water buffaloes at the Guangxi Buffalo Research Institute in China from 2000 to 2021. The sample comprised 1 local water buffalo (D), 46 hybrid water buffaloes (Z), 31 Nili-Ravi water buffaloes (N), and 42 Murrah water buffaloes (M). The investigation focused on four reproductive traits: calving interval (CI), calf birth weight (CBW), dam birth weight (BW), and age at first calving (FCA).

FCA represents the days from birth to the initial calving in water buffaloes, and CI signifies the days between successive calvings. CBW pertains to calf birth weight, while BW is the weight of the dam at birth.

### 2.2. Sample Collection and Sequencing

Blood samples from water buffalo were collected via tail vein puncture using a vacuum blood collector. Genomic DNA extraction from the blood was performed using the phenol/chloroform method, and the integrity and yield of the genomic DNA were evaluated through agarose gel electrophoresis.

The genomic DNA was utilized to create paired-end short-insert libraries (350 bp) and subsequently sequenced using the Illumina HiSeq X Ten system (Illumina, San Diego, CA, USA).

### 2.3. Alignments and Variant Identification

The clean reads were aligned to the reference genome (UOA_WB_1) using BWA-MEM with default settings. Subsequently, Samtools, Picard tools, and Genome Analysis Toolkit (GATK, version 4.0) were employed for single nucleotide polymorphism (SNP) detection. All SNPs underwent filtering through the “Variant Filtration” module of GATK, employing standard parameters as follows: Variants with Quality Depth (QD) < 2; FS (Phred-scaled *p*-value using Fisher’s exact test for strand bias detection) > 60; MQRankSum < −12.5; ReadPosRankSum < −8; MQ < 40.0; mean sequencing depth of variants (across all individuals) < 1/3× and > 3×; SOR > 3.0; maximum missing rate < 0.1; and SNPs were limited to two alleles.

### 2.4. Variation Filtering

Markers (SNPs/Indels) with allele number larger than 2, missing data greater than 50%, minor allele frequency lesser than 5%, and heterozygosity greater than 80% were removed, resulting in 691,729 SNPs and 53,761 Indels.

The remaining SNPs were used for population genetic analysis and all remaining markers (including SNPs and Indels) were used for GWAS.

### 2.5. Principal Component Analysis (PCA)

Principal component analysis (PCA) was performed based on filtered SNPs using GCTA [16] (v1.92.2) software.

### 2.6. Kinship Analysis

Pair-wise relationship matrix (kinship matrix) was calculated with all filtered SNPs using GCTA [16] (v1.92.2) software.

### 2.7. Phylogenetic Tree Construction

The neighbor-joining (NJ) tree based on filtered SNPs was constructed using MEGA-X [17] software v10.2 (p-distance model) with a total of 1000 bootstrap replicates.

### 2.8. Genome-Wide Association Mapping

Genome-wide association mapping was conducted in TASSEL [18] v.5.2.54, employing a mixed linear model (MLM(Q+K)) that incorporates both population structure and kinship as fixed and random effects, respectively. Population structure was addressed by utilizing the first three principal components, while kinship was considered using the kinship matrix generated through GCTA software. Significance thresholds were set at Bonferroni-adjusted genome-wide significance (0.05/745,490 = 6.71 × 10^−8^) and suggestive significance (5 × 10^−6^) to detect meaningful associations. Candidate genes (CAGs) located within the 50 kb region upstream or downstream of significantly associated markers were then identified.

## 3. Results

### 3.1. Phenotypic Parameters and Marker Information

We conducted a statistical analysis of four suggestive reproductive traits in 120 water buffaloes (summarized in Table 1). The phenotypic distributions of these traits are distinct and generally align with expected biological patterns. This alignment facilitates the identification of variant loci that exhibit significant differences in phenotypes, suggesting that nearby genes may play a role in controlling these traits. After performing quality control on SNP loci, we identified a total of 745,490 variant loci, including 691,729 SNPs and 53,761 Indels.

To enhance accuracy and minimize false positives, the GWAS analysis incorporates adjustments based on the genetic background and stratification of the study materials. Therefore, conducting a population structure analysis is essential, taking into account the filtered markers. This ensures that the model accurately reflects the genetic diversity and structure of the population, thereby improving the reliability of the GWAS results.

### 3.2. Population Structure Analysis

We generated a population structure plot using PCA (Figure 1). The plot illustrates clear stratification among the sample populations, suggesting potential population differences. After conducting PCA on the 120 buffalo samples, covering the four reproductive traits, the first three principal components (PCs) explained 4.45%, 4.39%, and 2.33% of the variation, collectively capturing around 11% of the total variation. The samples are distinctly grouped into three clusters, indicating significant population stratification. This underscores the presence of population differences among the sample groups, potentially leading to false positives in subsequent association analyses. Consequently, we incorporated the first three PCs as covariates in the analysis model to mitigate the risk of false positives and enhance result accuracy.

The results highlight significant familial correlations among some samples, necessitating further adjustments to the association analysis model. In unrelated groups or populations with unclear pedigrees, kinship represents the relative genetic similarity between two specific materials compared to the genetic similarity between any two materials. In our experiment, the heat map derived from the kinship matrix demonstrates evident familial correlations among specific samples (Figure 2). Consequently, we included the kinship matrix as a covariate in the analysis model to enhance result accuracy.

### 3.3. Results of the Genome-Wide Associations

GWAS was performed using logistic regression under an additive genetic model, incorporating five covariates: the first three principal components and the kinship matrix. A total of 52 SNPs were identified as significant (*P* < 5 × 10^−6^; two-tailed), as shown in the Manhattan plot (Figure 3) and the quantile–quantile (QQ) plot (Figure 4). Detailed information, including *p*-values, and percentage of variance explained (PVE), is provided in Table 2, which lists the SNPs within genes. Our research identified 52 suggestive regulatory loci associated with reproductive traits in water buffalo. Based on a 50 kb interval, we annotated these loci to 58 candidate genes. Notably, one significant SNP is located within the *GRM1* gene locus, and several gene loci contain two or more independent SNPs: *NCKAP5* contains 3 SNPs, *AGBL4* contains 2 SNPs, and *NRXN1* contains 2 SNPs.

To better visualize the associated loci, we generated a plot depicting the gene locations on chromosomes (Figure 5). A total of 58 genes linked to SNPs were identified across 15 chromosomes, with 14 of these genes containing SNPs within their genomic regions. Notably, the X chromosome exhibits sparse SNP density, while the highest SNP densities are observed on chromosomes 2, 8, and 13.

### 3.4. Kyoto Encyclopedia of Genes and Genomes Pathway Analysis of Candidate Genes

Through pathway analysis of these candidate genes, we found that they are significantly enriched in the FOXO signaling pathway, calcium ion pathways, estrogen pathways, phospholipase D signaling pathway, and glutamate metabolism. These signaling pathways are closely related to water buffalo reproductive traits either directly or indirectly (Figure 6). This suggests that the gene loci (*AGBL4*, *GRM1*, *NCKAP5*, *NRXN1*) may be important regulatory sites for controlling reproductive traits in water buffalo, providing strong support for improving future water buffalo reproduction rates.

## 4. Discussion

### 4.1. Control of Effective Parameters in GWAS Models

In our study, we used principal component analysis (PCA) results as covariates to genetically control genotype-by-environment (G×E) interactions. However, incorporating covariance from kinship relationships into the model can lead to overcorrection, reducing false positives but potentially overlooking loci with small effects. Despite this limitation, our study, which encompasses diverse water buffalo breeds and their hybrids, significantly enhances the efficacy of identifying genetic loci influencing phenotypes through GWAS. Importantly, we recognize that a well-characterized population structure is crucial for improving the reliability of data analysis. This aspect will be a key consideration in our future research efforts.

### 4.2. Suggestive Influencing Conditions for Water Buffalo Reproductive Traits and Insights from This Study

Previous research has established a significant link between reproductive traits and cow reproductive health. Researchers used classical repeatability and random regression models to analyze the variation in reproductive efficiency among 12,554 cows of the Retinta Spanish breed. They estimated deviations between optimal female performance and actual parity at different stages of the cows’ lives. Their analysis revealed that BTA4 and BTA28 significantly influence the reproductive efficiency of dairy cows [19,20].

Three of the four suggestive loci we identified are linked to the regulation of cow reproduction, while one is directly associated with sperm production and embryonic development. The most strongly associated locus is located on chromosome 10 in water buffalo within the gene *GRM1* (metabotropic glutamate receptor 1). This gene encodes mGluRs, which are typically found as dimers on cell membranes. Glutamate, an excitatory neurotransmitter, plays a role in synaptic modulation during reproduction by integrating signals from wakefulness and reproductive-promoting neuromodulators via Homer1a (a suggested reproductive regulatory factor) and mGluR1/5 (a subtype group of mGluRs) [21]. *NCKAP5* (NCK Associated Protein 5), predicted to be involved in microtubule formation and disassembly, is located on chromosome 2 in water buffalo, with three SNPs within its locus. Previous GWAS studies have associated *NCKAP5* with age at first calving (AFC). By analyzing 2400 genotyped animals, these studies identified 25 candidate genes for reproductive traits, including *CHD7, CLVS1, EVX2, MAT2B, NUDCD2, GPR39, NCKAP5, LYPD1, HOXD13, SEMA5B, CCNG1, SEMA5A, BRF1, PSEN2, CACHD1, SUGT1, ELF1, SNORA70, AKT1, TM2D1, SLF1, MCTP1, PABPC1, MTRF1*, and *ADCY2* [22]. Additionally, other studies have indicated that *NCKAP5* is a regulatory gene for one of the selection traits in Limousin cattle [23]. *AGBL4* (ATP/GTP-binding protein-like 4) is a metallo-carboxypeptidase involved in the deglutamylation process of proteins, particularly in the modification of tubulin. It is also associated with antiviral defense responses. The modification of tubulin is crucial for cell division, which is especially important in the generation of germ cells and early embryonic development [24]. Therefore, *AGBL4* may indirectly influence reproductive performance by regulating tubulin function.

### 4.3. Limitations and Future Directions

Our study aims to preliminarily investigate ways to enhance the reproductive performance of water buffalo and identify genetic loci associated with the age at first calving (AFC). However, we have identified several limiting factors that may affect the accuracy of our GWAS (genome-wide association study) results. Firstly, the small sample size limits the reliability of our findings, allowing us to provide only indicative rather than definitive conclusions [25,26,27]. Therefore, in future research, we plan to increase the sample size to address this limitation. Additionally, literature reviews suggest that factors such as photoperiod, breeding season, and parity also significantly impact AFC [28,29,30,31]. Our current study lacks data on these variables, which constitutes another limitation of our preliminary analysis. To better understand the mechanisms underlying reproductive traits in water buffalo, we aim to incorporate these variables into our subsequent studies [32,33]. We are committed to providing more robust data support and technical guidance for the development of water buffalo breeding programs.

## 5. Conclusions

In this study, our research identified 52 suggestive regulatory loci associated with reproductive traits (age at first calving (FCA)) in water buffalo. Based on a 50 kb interval, we annotated these loci to 58 candidate genes. These candidate genes are significantly enriched in the FOXO signaling pathway, estrogen pathways, and calcium ion signaling pathways. Among these, four suggestive genes (*AGBL4*, *GRM1*, *NCKAP5*, and *NRXN1*) were found to be involved in embryonic development and the regulation of mammalian reproduction.

These genes not only provide theoretical support for understanding water buffalo reproduction but also offer potential target genes for practical breeding programs. In summary, by identifying QTLs associated with water buffalo reproductive traits, our study enhances the understanding of the reproductive mechanisms in water buffalo and provides scientific basis and technical support for future genetic breeding strategies.

## Figures and Tables

**Figure 1 genes-16-00422-f001:**
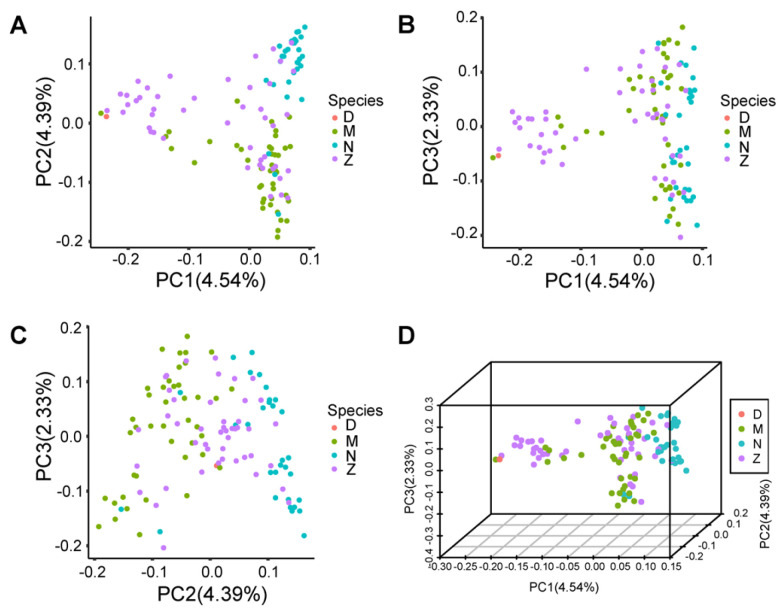
Principal component analysis (PCA) results based on filtered SNP markers. The PCA analysis was conducted using filtered SNP markers to obtain the variance explained by each principal component (PC) and the score matrix of samples on each PC. The population structure is visualized through pairwise scatter plots (**A**–**C**) and a 3D plot (**D**) of the top three principal components. The colored circles represent four populations: D, M, N, and Z, each comprising 1 local buffalo, 42 Murrah buffalo, 31 Nili-Ravi buffalo, and 46 hybrid buffalo. The axes of the plots represent the scores of each PC, with the percentage values in parentheses indicating the variance explained by each PC, representing the proportion of total variance contributed by that PC in relation to all PCs. Orange: Represents group D. Green: Represents group M. Blue: Represents group N. Purple: Represents group Z.

**Figure 2 genes-16-00422-f002:**
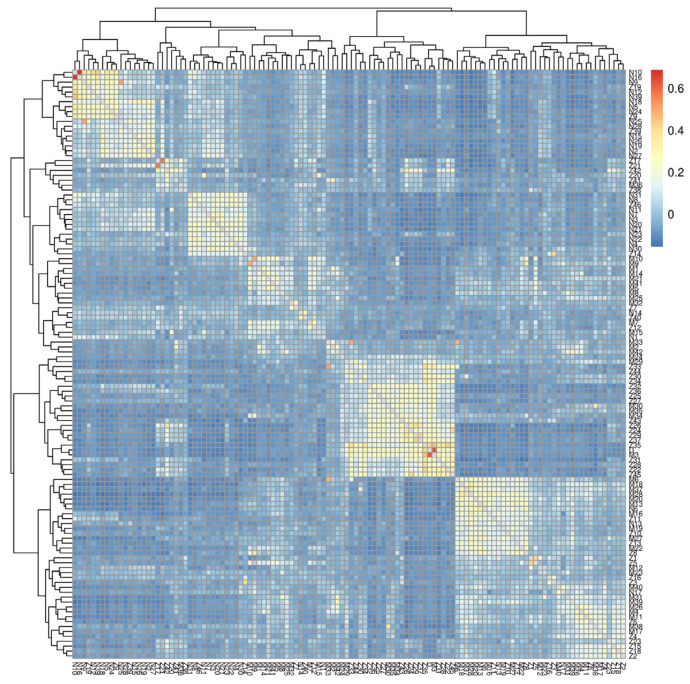
Heatmaps and neighbor-joining phylogenetic tree (NJ) of kinship among samples. Using filtered markers, we conducted kinship analysis with the GCTA software (v1.92.2) to obtain the kinship matrix for pairwise relationships among samples. A heatmap was generated based on this kinship matrix, where the color gradient from blue to red signifies increasing proximity in kinship relationships. Furthermore, a neighbor-joining phylogenetic tree was constructed using MEGA-X software (model: p-distance; bootstrap: 1000 replicates). The cattle populations D, M, N, and Z represent local buffalo, Murrah buffalo, Nili-Ravi buffalo, and hybrid buffalo, respectively.

**Figure 3 genes-16-00422-f003:**
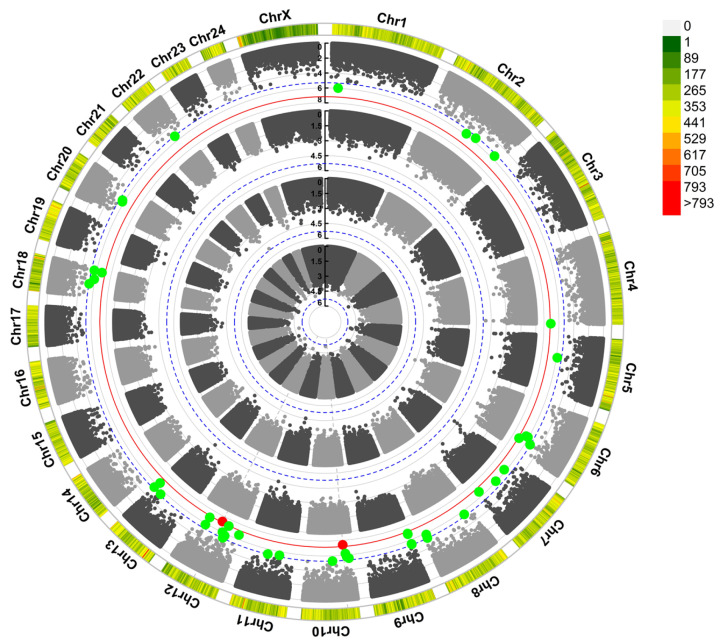
Manhattan plots illustrate GWAS results for reproductive traits. The x-axis represents chromosomal positions, while the y-axis displays -log_10_
*p*-values indicating allele-phenotype associations. The blue dashed line signifies the suggestive significance cutoff, with green spots denoting SNPs reaching suggestive significance. The red line corresponds to the Bonferroni thresholds (*p* = 0.05), and red spots indicate SNPs with significance. Traits, arranged from inner to outer lanes, include calving interval (CI), calf birth weight (CBW), dam birth weight (BW), and age at first calving (FCA). Green dots represent SNPs meeting the suggested threshold. Red dots represent SNPs meeting the significance threshold. The color scale in the top right indicates SNP density on the chromosome, with green showing low density and red showing high density.

**Figure 4 genes-16-00422-f004:**
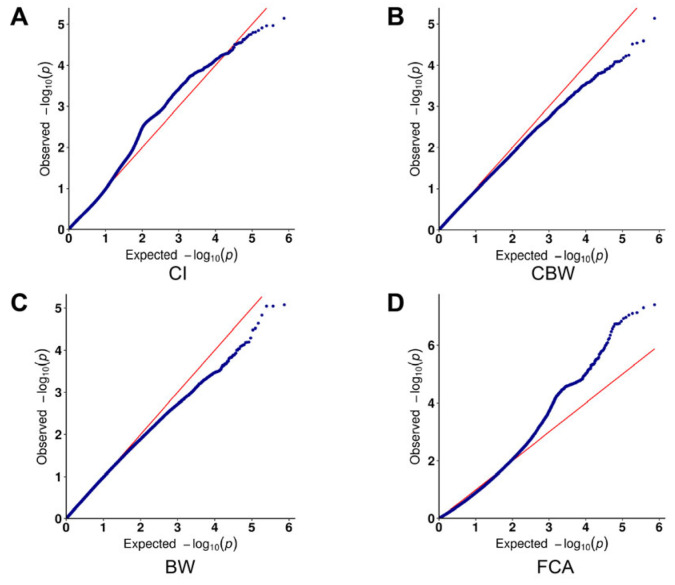
Quantile–Quantile (QQ) Plots. The QQ plots illustrate the negative logarithms of expected *p*-values on the x-axis and observed *p*-values on the y-axis (right panel). (**A**) calving interval (CI), (**B**) calf birth weight (CBW), (**C**) dam birth weight (BW), and (**D**) age at first calving (FCA).

**Figure 5 genes-16-00422-f005:**
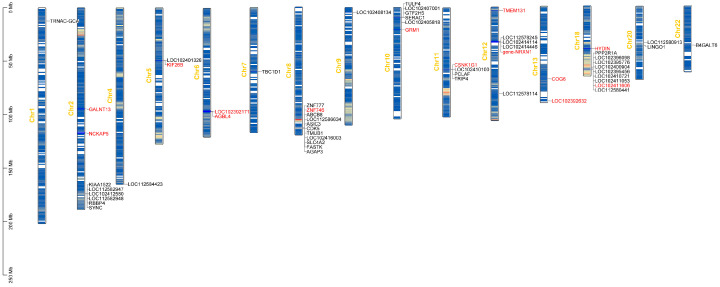
Plot of significant genes on chromosomes. The gene density heatmap visually represents the distribution of genes, ranging from sparse (blue) to dense (red). Red gene symbols indicate the presence of SNPs within the genes, while others are located within 50 kb upstream and downstream of the SNP. In total, 58 genes associated with SNPs were identified on 15 chromosomes. These genes are the candidate genes identified in the analysis.

**Figure 6 genes-16-00422-f006:**
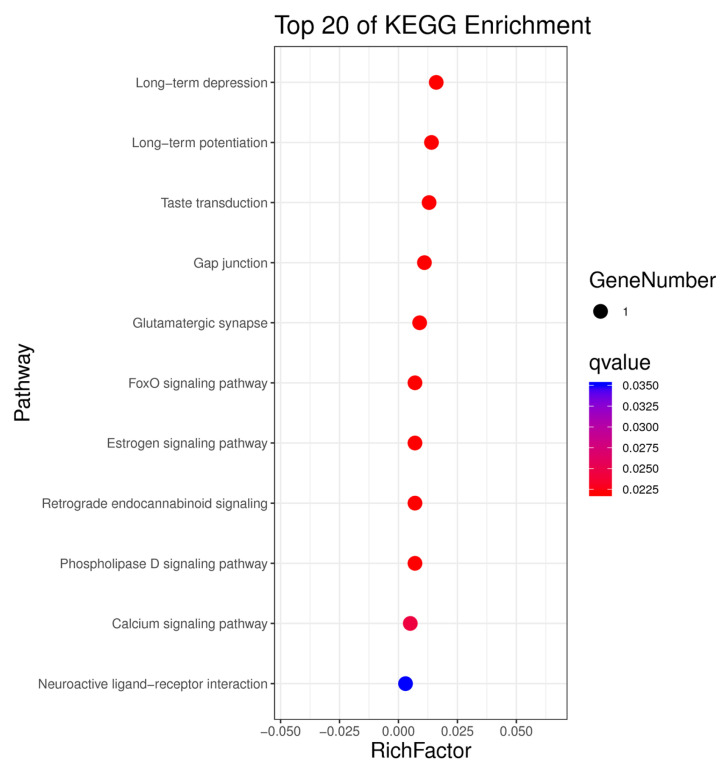
Bubble plots for KEGG enrichment analysis. The y-axis denotes the entry name, while the x-axis’s rich factor signifies the ratio of listed genes within the entry to the total genes constituting it. The size of each filled circle corresponds to the number of candidate genes in the entry, and the color indicates the saliency of the entry.

**Table 1 genes-16-00422-t001:** Statistical description of study traits *.

Traits	Records	Mean	SD	Min	Max
CI	51	581.68 (Days)	194.21	414	1336
CBW	106	35.02 (Months)	4.89	18.5	46
BW	115	35.47 (Months)	5.16	20	46
FCA	105	44.41 (Months)	15.50	5.5	120.5

* CI, calving interval; CBW, calf birth weight; BW, dam birth weight; FCA, age at first calving. Calving interval (CI) is measured in days. Calf birth weight (CBW), dam birth weight (BW), and age at first calving (FCA) are measured in months.

**Table 2 genes-16-00422-t002:** Genome-wide significant SNPs associated with the trait of age at first calving *.

SNP	Chr	Position(bp)	Candidate Gene	*p*-Value	PVE
1	1	16038140	*TRNAC-GCA*	1.06 × 10^−6^	0.51
2	2	94758634	*GALNT13*	1.54 × 10^−6^	0.50
3	2	116842309	*NCKAP5*	3.08 × 10^−6^	0.37
4	2	116842310	*NCKAP5*	3.08 × 10^−6^	0.37
5	2	116842325	*NCKAP5*	3.08 × 10^−6^	0.37
6	2	173941889	*LOC102412580,KIAA1522,* *SYNC,LOC112582948,* *RBBP4*	2.62 × 10^−6^	0.36
7	4	164836341	*LOC112584423*	8.04 × 10^−8^	0.53
8	5	49042788	*LOC102401328* *,* *KIF26B*	1.37 × 10^−6^	0.36
9	6	84593264	.	8.09 × 10^−7^	0.55
10	6	84593519	.	4.45 × 10^−7^	0.54
11	6	96791388	*LOC102392171* *,* *AGBL4* *,*	9.2 × 10^−8^	0.51
12	6	96811758	*LOC102392171*	4.69 × 10^−6^	0.33
12	6	96811758	*AGBL4*	4.69 × 10^−6^	0.33
13	7	28031213	.	5.72 × 10^−7^	0.45
14	7	59963243	*TBC1D1*	8.62 × 10^−7^	0.42
15	7	103706036	.	2.29 × 10^−7^	0.60
16	8	17664369	.	2.95 × 10^−6^	0.59
17	8	112491377	*ZNF746* *,* *ZNF777*	1.58 × 10^−6^	0.32
18	8	113607118	*LOC112586634,ASIC3,* *LOC102416003,FASTK,* *TMUB1,ABCB8,* *CDK5,SLC4A2,* *AGAP3*	4.77 × 10^−6^	0.31
19	9	4404838	*LOC102408134*	1.51 × 10^−7^	0.51
20	9	4404850	*LOC102408134*	1.21 × 10^−7^	0.51
21	9	4404918	*LOC102408134*	3.6 × 10^−6^	0.38
22	9	4404920	*LOC102408134*	4.29 × 10^−6^	0.37
23	10	9287335	*LOC102407001,GTF2H5,* *SERAC1*	2.86 × 10^−6^	0.31
24	10	13862680	*LOC102405818*	1.76 × 10^−6^	0.42
25	10	16312432	.	6.3 × 10^−7^	0.37
26	10	20660384	*GRM1*	4 × 10^−8^	0.50
27	10	45617271	.	4.97 × 10^−6^	0.35
28	10	45617310	.	4.97 × 10^−6^	0.35
29	11	32169042	.	3.18 × 10^−6^	0.34
30	11	57697044	*PCLAF,LOC102410100,* *TRIP4,CSNK1G1*	4.69 × 10^−6^	0.33
31	12	3176008	*TMEM131*	2.72 × 10^−7^	0.54
32	12	31988278	*LOC112578245* *,* *NRXN1*	1.92 × 10^−6^	0.50
33	12	32801224	*LOC102414114,LOC102414446,* *NRXN1*	7.54 × 10^−8^	0.54
34	12	34019714	.	4.38 × 10^−6^	0.38
35	12	39801133	.	9.78 × 10^−7^	0.61
36	12	50669172	.	5.07 × 10^−8^	0.43
37	12	81013839	*LOC112578114*	1.11 × 10^−7^	0.49
38	12	81118534	.	1.49 × 10^−6^	0.35
39	13	67984517	*COG6*	3.56 × 10^−6^	0.42
40	13	86238940	.	3.38 × 10^−7^	0.54
41	13	88603805	*LOC102392632*	2.9 × 10^−6^	0.40
42	18	28221520	.	4.57 × 10^−6^	0.37
43	18	28221526	.	4.57 × 10^−6^	0.37
44	18	28221527	.	4.57 × 10^−6^	0.37
45	18	39920231	*HYDIN*	1.22 × 10^−6^	0.44
46	18	58127554	*LOC102396098,LOC102395776,* *LOC102400904,LOC102395456,* *PPP2R1A*	1.86 × 10^−7^	0.52
47	18	58127556	*LOC102396098,LOC102395776,* *LOC102400904,LOC102395456,* *PPP2R1A*	1.86 × 10^−7^	0.52
48	18	58127575	*LOC102396098,LOC102395776,* *LOC102400904,LOC102395456,* *PPP2R1A*	1.78 × 10^−7^	0.52
49	18	58127583	*LOC102396098,LOC102395776,* *LOC102400904,LOC102395456,* *PPP2R1A*	1.87 × 10^−7^	0.52
50	18	59367852	*LOC112580441,LOC102410721,* *LOC102411053,LOC102411606*	2.08 × 10^−6^	0.41
51	20	33712610	*LOC112580913*	1.63 × 10^−6^	0.43
52	20	36725181	*LINGO1,B4GALT6*	2.05 × 10^−6^	0.32

* PVE, percentage of variance explained.

## Data Availability

Study content data may be obtained by contacting the corresponding author.

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
