# Peer review of "Genome-Wide Association Study Identifies Potential Regulatory Loci and Pathways Related to Buffalo Reproductive Traits"

_genes, 2025, doi:10.3390/genes16040422_

Round 1

Reviewer 1 Report

Comments and Suggestions for Authors

The manuscript entitled “Genome-wide association study reveals key regulatory loci and pathways for enhanced buffalo reproductive performance” purports to describe loci that influence reproductive traits in water buffalo.  While extensive in the number of cows genotyped by association mapping, there are a number of substantial concerns with the manuscript as presented.

  1. the authors repeatedly call the associations identified as reflecting “key” regulatory loci. This is simply not true based on their data. These cannot be defined as “key”. Furthermore, there were only two significant chromosomal regions associated with a single reproductive trait of the four evaluated. The vast majority of the associations were suggestively associated with age at first calving interval and no other trait demonstrated associations. Additionally, the variation captured by the principal components is modest and that too does not indicate they are “key” associations.
  2. Thus, the title and much of text is misleading when the authors claim these are regulatory loci for reproductive performance writ large. The association is only for age at first calving.
  3. The authors did not detail the phenotypic criteria in the methods. What was the body condition score of the cows at breeding. Body condition can significantly influence reproductive traits most especially age at first calving which reflects entrance to puberty of the female.  Additionally, the reproductive cycle of water buffalo is photoperiod sensitive. How was that accounted for in the author’s study? If that wasn’t accounted for the significant loci may actually be associated with photoperiod response and not reproduction per se.
  4. Using a single individual as representing “a population” is not appropriate. The Z population is a single animal. They should omit that animal completely from the analysis.
  5. The data need to be confirmed in an independent cohort of animals before making these broad claims of “key” loci. The authors themselves imply that this is necessary in lines 179-184.
  6. Did the authors do a more limited pathway analyses on only those loci that reached a certain significance threshold rather than the many that were merely suggestive?
  7. Table 2 legend states “Genome-wide significant SNPS associated …” whereas really there are only a few that reach genome wide significance. The legend is misleading. In that table, those that are truly significant should be distinguished from the others with bolding or some other mechanism to indicate the significance. Also there is no reason to label the SNPs 1-52 in table 2 (and why does the abstract, text, and conclusion state 58?).
  8. The introduction focuses on the need for sequencing and the costs associated (paragraphs lines 50-71) yet there is no real discussion of that in the results or the discussion. The emphasis of sequencing in the introduction does not make sense and the inclusions seems to be more relevant to a different paper. The introduction should focus on the topic at hand: reproduction and provide context in what other investigators have found in bovine species.
  9. Similar to the comment above about focusing the introduction on bovine species reproduction, the discussion needs to focus on bovine, as opposed to using the human as a comparator. There are many published studies on loci involved in the reproductive pathway in bovine as well as other domesticated ungulates which would be a more appropriate comparison and provide better context to the scientific field.
  10. Figures and table legends need improvement. Units are missing, there are colors in some figures without an explanation of their meaning (for example green dots and color spectrum on figure 4) and there are vague titles (For example figure 6, plot of significant genes on chromosomes---are these the truly significant genes associated with age at first calving interval?).
  11. The first paragraph of the discussion has little value or relationship to this paper. It is also unclear the importance of what the authors are intending to convey in the sentence on lines 272-273 (Altered reproductive patterns can negatively impact reproductive function, resulting in compromised reproductive outcomes) because this statement is self-evident. Perhaps there is something missing?
  12. Lines 293-294; lines 299-301 something seems to be missing.
  13. Line 368-370 is not accurate. The data only demonstrate significant association with age at first calving (and without having body condition score or accounting for photoperiod even that association is suspect).

Author Response

Thank you very much for your valuable feedback. Your suggestions have greatly improved the research prospects and overall quality of our manuscript. We have carefully addressed each of your comments and have made the necessary revisions. Our detailed responses to your feedback are provided in the attached document. We appreciate your attention to this matter.

Reviewer 2 Report

Comments and Suggestions for Authors

Manuscript submitted for review: "Genome-Wide Association Study Reveals Key Regulatory Loci and Pathways for Enhanced Buffalo Reproductive Performance" A genetic study was conducted to identify the genes responsible for reproductive performance in buffaloes. Considering that buffaloes are animals that have difficulty adapting to intensive farming and often have fertility problems during artificial insemination, the study is very relevant. In addition to the relevance of the topic, the fact that buffaloes are late-maturing animals (it takes a long time to reach breeding maturity), with a long gestation period and much more difficult to synchronize estrus and artificial insemination, compared to cattle, I believe that the topic is very relevant and related to the practice of improving buffalo breeding.

The study was conducted with local, hybrid and two of the most common buffalo breeds in the world Nili-Ravi and Murrah. By selecting breeds, the authors have tried to find a relationship between the genes sought in the study and the buffalo breeds. It is well known that in buffaloes, the less productive and primitive breeds tend to have better reproductive performance, but only under natural conditions. In the case of highly productive cultural breeds such as Nili-Ravi and Murrah, they are much better amenable to synchronization and artificial insemination. In this line of thought, I believe that the authors have very correctly selected the object of their study and I have no comments.

The results are well presented with the exception of table and figure 1, where in my opinion only table 1 should remain. A detailed and accurate discussion has been made with the findings of other authors on the subject and the results obtained have been compared. The conclusion drawn stems from the results obtained.

Author Response

(The authors gave the same response as above.)

Reviewer 3 Report

Comments and Suggestions for Authors

This study takes advantage of whole genome sequencing and genome-wide association approaches to unravel potential molecular markers that could predict fertility in water buffaloes.  Based on a tremendous potential in understanding reproduction on a molecular level because of genomics and the lack of data on water buffaloes as an important cattle species in certain parts of the world, this study has merit and could serve as an important source of information for the relevant researchers.

The experiments follow a standard approach and apply appropriate models to reach the goals set. It is a shame though that neither semen analysis or the levels of reproductive hormones were not included into the data analysis since these could add more insights to the outcomes of this paper. Also, breed-related differences could be an interesting aspect since different breeds were included in the study.

Finally the authors could critically address any limitations that may have impacted the outcomes of the study as well as indicate future directions or strategies in their research.

Author Response

(The authors gave the same response as above.)

Reviewer 4 Report

Comments and Suggestions for Authors

Dear Authors, the topic presented by you is quite interesting, in light of the identification of 58 loci significantly associated with different key reproductive traits. The study design is valid as well as the general overall exploration of key regulators loci and pathways for enhanced buffalo reproductive performance by means of genome wide association study (GWAS) and statistical tests related, as Principal Component Analysis.

The introduction is complete such as the materials and methods, that are well detailed. The section that has to be modified is the discussion, too long and dispersive, because only the last paragraph clearly reports the results of your study. Outcomes have to be deeply discussed, comparing them with literaure data on the topic.

Please, adequate the references to the format of the "Genes" Journal, folowing the guidelines.

Comments on the Quality of English Language

English language has to be revised to make the text more fluent. There are also formatting and dotting errors.

Author Response

(The authors gave the same response as above.)

Round 2

Reviewer 1 Report

Comments and Suggestions for Authors

This reviewer appreciates all the revisions provided by the authors. Yet the fatal flaw in the research design remains.  By not accounting for the phenotypes of the cows the data cannot be attributed to the reproductive traits described and thus do not extend the literature in an impactful manner.